# A selectivity filter at the intracellular end of the acid-sensing ion channel pore

Timothy Lynagh[1], Emelie Flood[2], Céline Boiteux[2], Matthias Wulf[1], Vitaly V Komnatnyy[1], Janne M Colding[1], Toby W Allen[2], Stephan A Pless[1]*

[1]Center for Biopharmaceuticals, Department of Drug Design and Pharmacology, University of Copenhagen, Copenhagen, Denmark; [2]School of Science, RMIT University, Melbourne, Australia

**Abstract** Increased extracellular proton concentrations during neurotransmission are converted to excitatory sodium influx by acid-sensing ion channels (ASICs). 10-fold sodium/potassium selectivity in ASICs has long been attributed to a central constriction in the channel pore, but experimental verification is lacking due to the sensitivity of this structure to conventional manipulations. Here, we explored the basis for ion selectivity by incorporating unnatural amino acids into the channel, engineering channel stoichiometry and performing free energy simulations. We observed no preference for sodium at the "GAS belt" in the central constriction. Instead, we identified a band of glutamate and aspartate side chains at the lower end of the pore that enables preferential sodium conduction.

*For correspondence: stephan. pless@sund.ku.dk

Competing interests: The authors declare that no competing interests exist.

## Introduction

Changes in extracellular environment and chemical signals from adjacent cells are rapidly converted into electrical signals by ligand-gated ion channels in the cell membrane. This is exemplified by acid-sensing ion channels (ASICs) that mediate excitatory sodium ($Na^+$) influx in response to increases in extracellular proton concentration during neurotransmission (*Du et al., 2014*; *Kreple et al., 2014*). The prototypical example, ASIC1a, is widely expressed in the mammalian nervous system, notably the brain, where it is involved in learning, memory and anxiety (*Pidoplichko et al., 2014*; *Wemmie et al., 2002*; *Ziemann et al., 2009*) and neuroinflammation and ischemia (*Voilley et al., 2001*; *Xiong et al., 2004*). In central and peripheral neurons, ASIC1a, in concert with ASIC2a and ASIC3, mediates pain and nociception, drawing much attention as a pharmacological target for novel analgesics (*Baron and Lingueglia, 2015*).

   ASICs are excitatory because of approximately 10-fold selectivity for $Na^+$ over potassium ($K^+$; [*Gründer and Pusch, 2015*; *Waldmann et al., 1997*]), a feature conserved in most vertebrate members of the overarching DEG/ENaC family of channels (named after invertebrate d̲egenerins and ver̲tebrate e̲pithelial N̲a̲+ channels). Experiments on heteromeric ENaCs have shown effects on ion selectivity and greater permeation of larger cations upon mutation of a highly conserved G-X-S motif in the pore-forming helix, suggestive of a size exclusion filter (*Kellenberger et al., 1999a*; *Snyder et al., 1999*). By homology, this idea naturally extended to ASICs (*Kellenberger and Schild, 2002*), receiving recent support from the presence of a constriction half-way down the pore in an X-ray structure of chick ASIC1 in a putative open-channel state (*Baconguis et al., 2014*). In ASIC1a, the constriction is formed by G-A-S motifs from each of three adjacent subunits, suggesting that this 'GAS belt' forms the ASIC selectivity filter, presumably favoring $Na^+$ by excluding larger $K^+$ ions (*Baconguis et al., 2014*). Unfortunately, the role of the GAS belt in ion selectivity lacks experimental verification, as the mutation of presumed critical G10' and S12' residues (see *Figure 1—figure supplement 1* for numbering) renders homomeric ASICs non-functional (*Carattino and Della Vecchia,*

*2012*; *Li et al., 2011b*; *Yang et al., 2009*). Moreover, selective $Na^+$ conduction has been ascribed to G10' main chain carbonyl oxygen atoms that point into the pore (*Baconguis et al., 2014*), and the role of main chain atoms cannot be addressed by conventional mutagenesis. Thus, the molecular determinants of ion selectivity and in turn the basis for excitatory function in this family of ligand-gated ion channel remains an open question.

Here, we have replaced the main chain amide carbonyl of G10' with an ester carbonyl in ASIC1a, which unlike conventional mutagenesis, allowed us to measure function in channels with alterations to the G10' main chain oxygen. We also ran molecular dynamics (MD) – free energy simulations using recent high resolution X-ray structures (*Baconguis et al., 2014*; *Jasti et al., 2007*), allowing us to assess the contribution of pore-lining residues to $Na^+$ and $K^+$ permeation. Both approaches support the conclusion that other parts of the pore contribute more markedly to ion selectivity than the GAS belt. Via extensive mutagenesis, further unnatural amino acid incorporation and engineered subunit stoichiometry, we established that a band of negatively charged side chains at the lower end of the channel pore enables preferential $Na^+$ conductance, revealing the molecular basis for excitatory ASIC1a function.

## Results

### Amide-ester mutation enables experimental test of G10' contribution to $Na^+$ conduction

We used in vivo nonsense suppression (*Dougherty and Van Arnam, 2014*) to replace A11' with its α-hydroxy analogue ('α') in ASIC1a channels expressed in *Xenopus laevis* oocytes (*Figure 1A,B*). This replaces the G10' main chain amide carbonyl with an ester carbonyl, decreasing the backbone dipole and thus the electrostatic surface potential near the G10' carbonyl oxygen (*Lu et al., 2001*). Unlike substitution via conventional mutagenesis of the GAS belt in ASICs, this substitution had remarkably little effect on general channel function, evident in unaltered proton-gated currents at A11'α channels (*Figure 1C*). To test if ion conduction was affected, we pulled outside-out patches and measured single channel $Na^+$ currents through A11'α channels and observed that $Na^+$ conductance was indistinguishable from wild-type (WT), despite the significant alteration of the G10' carbonyl oxygen (*Figure 1D*). Ion selectivity was assessed by measuring relative permeabilities of $Li^+$, $K^+$ and $Cs^+$ (*Figure 1—figure supplement 1*). A11'α channels maintained high, WT-like levels of $Na^+$ selectivity over the larger $K^+$ and $Cs^+$ (*Figure 1E*), with an *increase* in $P_{Na+}/P_{K+}$ that signifies interactions with G10' specific to $K^+$ conductance (addressed below).

### No preference for $Na^+$ ions at the GAS belt

We next explored $Na^+$ and $K^+$ permeation using MD simulations, based on the apparently open-channel ASIC1 structure (PDB:4NTW (*Baconguis et al., 2014*); *Figure 2A*). We observed no free energy preference for $Na^+$ ions at the GAS belt (*Figure 2B*), which was confirmed by independent calculations of relative binding free energy at this part of the channel (*Figure 2—figure supplement 1*). When passing the GAS belt, both $Na^+$ and $K^+$ maintain their solvation shell, but $K^+$ ions do so with more contribution from the protein, interacting with G10' amide carbonyl oxygen atoms (*Figure 2—figure supplement 2*). These interactions explain the lack of significant energetic difference between species and show why experimental alteration of G10' specifically affected $K^+$ permeability. Free energy differences between $Na^+$ and $K^+$ were observed at L7' and E18'-D21' (*Figure 2B*). These results are explained by partial dehydration of $K^+$ without sufficient compensation by backbone carbonyls around L7' (*Figure 2—figure supplement 2*) and high field strength binding (*Hille, 1972*) of $Na^+$ to negative charge at E18'-D21'. Furthermore, independent binding calculations have demonstrated a relative $Na^+/K^+$ stability of $1.4 \pm 0.1$ kcal/mol for binding at E18' (*Figure 2—figure supplement 1*). Simulations with NaCl or KCl also revealed formation of complexes with 1–2 ions and 1–2 carboxylate groups, involving E18' and D21' (*Figure 2C*). Although single ion occupancy dominates for both $Na^+$ and $K^+$, double ion occupancy occurred more frequently for $Na^+$ (*Figure 2C*) and this doubly occupied site was $3.0 \pm 0.4$ kcal/mol more stable for $Na^+$ than for $K^+$ (*Figure 2—figure supplement 1*). Thus, simulations identify E18'/D21' as a likely site to increase $Na^+$ concentration in a selective multi-ion permeation mechanism.

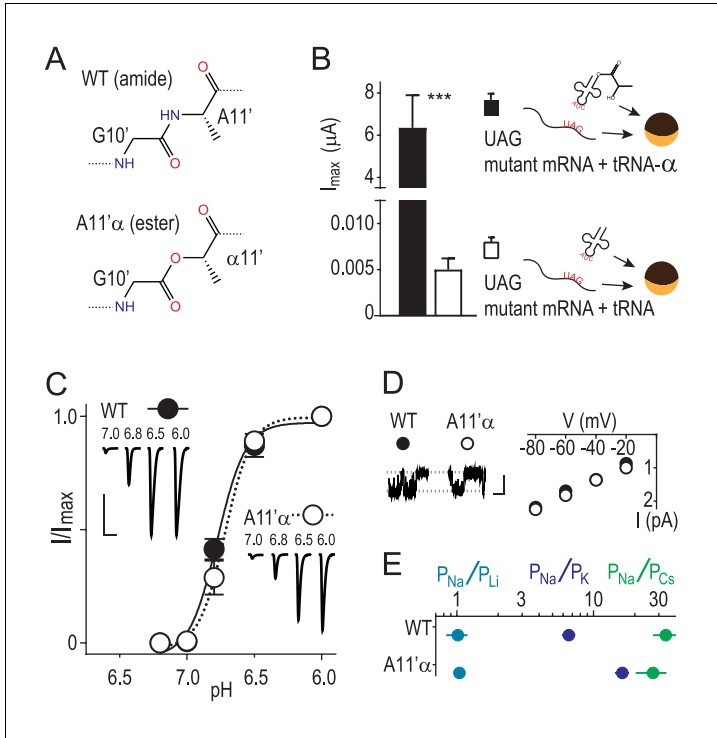

**Figure 1.** Amide-ester substitution to probe contribution of G10′ to Na$^+$ conduction. (**A**) A11′α substitution. (**B**) Successful incorporation of lactic acid ('α') into position A11′, indicated by large proton-gated currents (pH 6.0) at oocytes injected with A11′UAG mRNA and tRNA-α (Mean ± SEM, n = 6) but not with A11′UAG mRNA and tRNA without α (n = 7). ***p=0.001 (Student's t-test). (**C**) A11′α channels (pH$_{50}$ = 6.7 ± 0.03, n = 5) respond to increasing proton concentrations much like WT channels (pH$_{50}$ = 6.8 ± 0.02, n = 4; p=0.07, unpaired t-test). (**D**) Single channel Na$^+$ currents (scale bars: x, 50 ms; y, 2 pA) and mean current amplitude at different voltages (± SEM, n = 5–8). (**E**) Relative permeability ratios at WT and A11′α channels (mean ± SEM, n = 4–9; see *Figure 1—figure supplement 1*.).

The following figure supplement is available for figure 1:

**Figure supplement 1.** Numbering scheme for M2 residues.

## Selective ion binding is strongly determined by lower pore structure

These results demonstrate that the lower ASIC1a pore is able to provide substantial thermodynamic preference for Na$^+$ ions. However, the lower pore of this apparently open chick ASIC1/snake toxin complex (PDB:4NTW) is noticeably splayed (*Figure 3A*), possibly as a consequence of the removal of intracellular N- and C-termini for crystallization (*Baconguis et al., 2014*). We note that parts of these intracellular regions help form the conducting pore (*Pfister et al., 2006*) and affect efficiency of channel opening (*Jing et al., 2013*; *Wu et al., 2016*) in ASICs and also make crucial contributions to gating and ion conduction in other trimeric ion channels (*Gründer et al., 1999*; *Mansoor et al., 2016*). Indeed, a narrower lower pore than that seen in PDB:4NTW would resemble the selectivity filter of voltage-gated Na$^+$ channels (*Figure 3A*), in which cooperative Na$^+$ binding across subunits is known to contribute to ion selectivity (*Boiteux et al., 2014*; *Payandeh et al., 2011*). In the absence of such an ASIC structure, we examined ion binding to PDB:2QTS (*Jasti et al., 2007*), a closed-channel chick ASIC1 X-ray structure which resolves an additional 10 lower pore residues and shows a significantly narrower lower channel pore (*Figure 3B*). Although these simulations cannot inform on conduction, they allowed us to explore Na$^+$ and K$^+$ interactions in a pore for which M2 helices are less splayed. These simulations revealed preferential binding of a single Na$^+$ ion by up to 2.6 ± 0.3 kcal/mol near E18′ (*Figure 3B*; *Figure 2—figure supplement 1*). In this narrower pore structure, we observed predominant formation of double ion complexes with E18′ and/or D21′ from neighboring

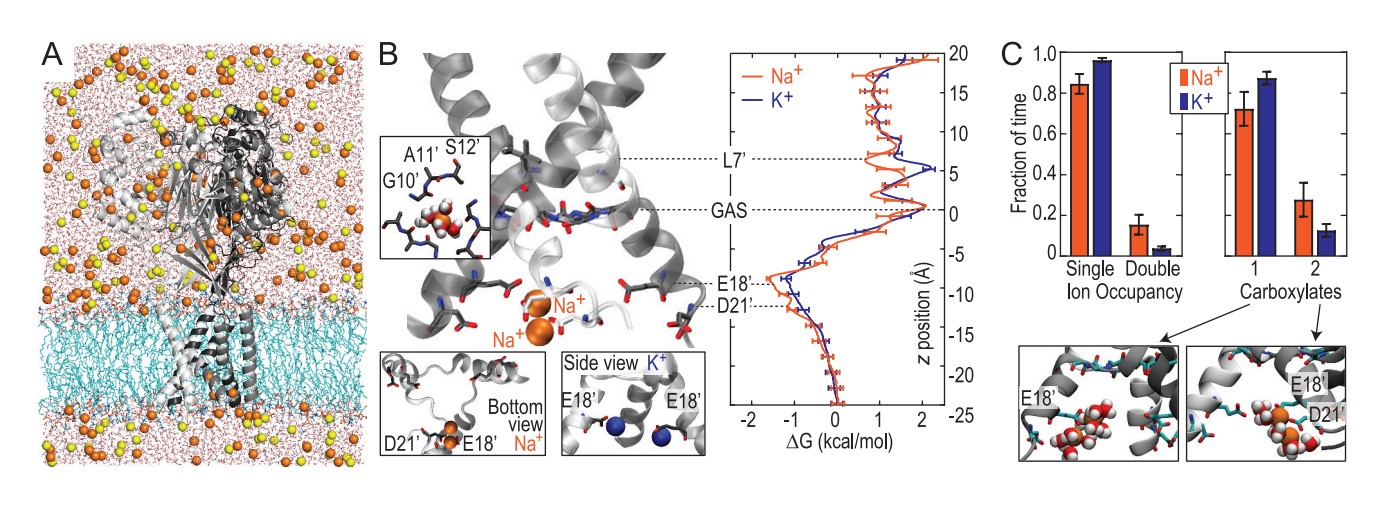

**Figure 2.** Free energy simulations with PDB:4NTW. (**A**) MD simulation system for PDB:4NTW (gray) embedded in a lipid bilayer (cyan) in NaCl solution (water, red and white sticks; Na+ and Cl-, orange and yellow balls). (**B**) Snapshots from simulation (left) and single ion-free energy profiles (right) for Na+ or K+ permeating the chick ASIC1 pore. See *Figure 2—figure supplement 1* for convergence analysis. Insets show hydrated Na+ in the GAS belt (left) and two Na+ ions or K+ ions at the level of E18' and D21' (lower). (**C**) Distribution of ion occupancies in sites formed by E18' and D21' in 4NTW. Single ion occupancy is favored for Na+ and K+, but double ion occupancy occurs more frequently for Na+ (left graph). For the doubly occupied site (right graph), binding to E18' dominates ('1 carboxylate'), but with increased probability of tight binding to both E18' and D21' ('2 carboxylates') for Na+. Insets: representative double Na+ ion complexes.

The following figure supplements are available for figure 2:

**Figure supplement 1.** Convergence of Free Energy Calculations.

**Figure supplement 2.** Ion coordination analysis.

subunits (*Figure 3C*, left graph), with a greater proportion of tightly bound complexes for Na+ (right graph). Such cooperative complex formation leads to a significant $5.9 \pm 0.6$ kcal/mol preference (*Figure 2—figure supplement 1*), aggressively favoring Na+ ions. Thus, although permeation is not observed in simulations with the closed-channel 2QTS, the closer proximity of adjacent helices provides a means for carboxylate side chains to exert substantial control on selective binding at the channel opening to promote efficient multi-ion Na+ conduction.

## E18' determines selective Na+ conductance

Our results suggest that GAS exerts less control over selectivity than previously thought and implicate other locations in selectivity. We therefore measured relative ion permeabilities in a series of mutants involving 16 additional pore-forming positions and found that L7', L14', E18' and D21' were the only residues whose mutation markedly altered ion selectivity (*Figure 4A*). This is consistent with the energetics for L7' and E18'/D21' in *Figure 2B*, although the influence of L14' has possibly been suppressed by the wide 4NTW vestibule. Regarding L7', it was only substantial changes in side chain size/length that dramatically altered selectivity, with L7'I and even L7'V substitutions not decreasing Na+/K+ selectivity (*Figure 4—figure supplement 1*). The fact that both E18'Q and E18'D mutations abolished selectivity (*Figure 4A*) suggests that both side chain charge and length make important contributions to Na+ conduction at this position. To test the role of charge, often obscured by the additional chemical changes in conventional acid-amide substitutions (*Pless et al., 2011*), we replaced glutamate with the isosteric but uncharged analogue 4-nitro-2-aminobutyric acid ('€', *Figure 4B*). Although currents at oocytes expressing E18'€ channels were relatively small (*Figure 4—figure supplement 1*), we could measure approximate reversal potentials for Na+ and K+ and establish that $P_{Na+}/P_{K+}$ was reduced to unity for E18'€ (*Figure 4B*), thus verifying that E18' negative charge is essential for Na+ selectivity. Moreover, single channel activity through E18'D channels was

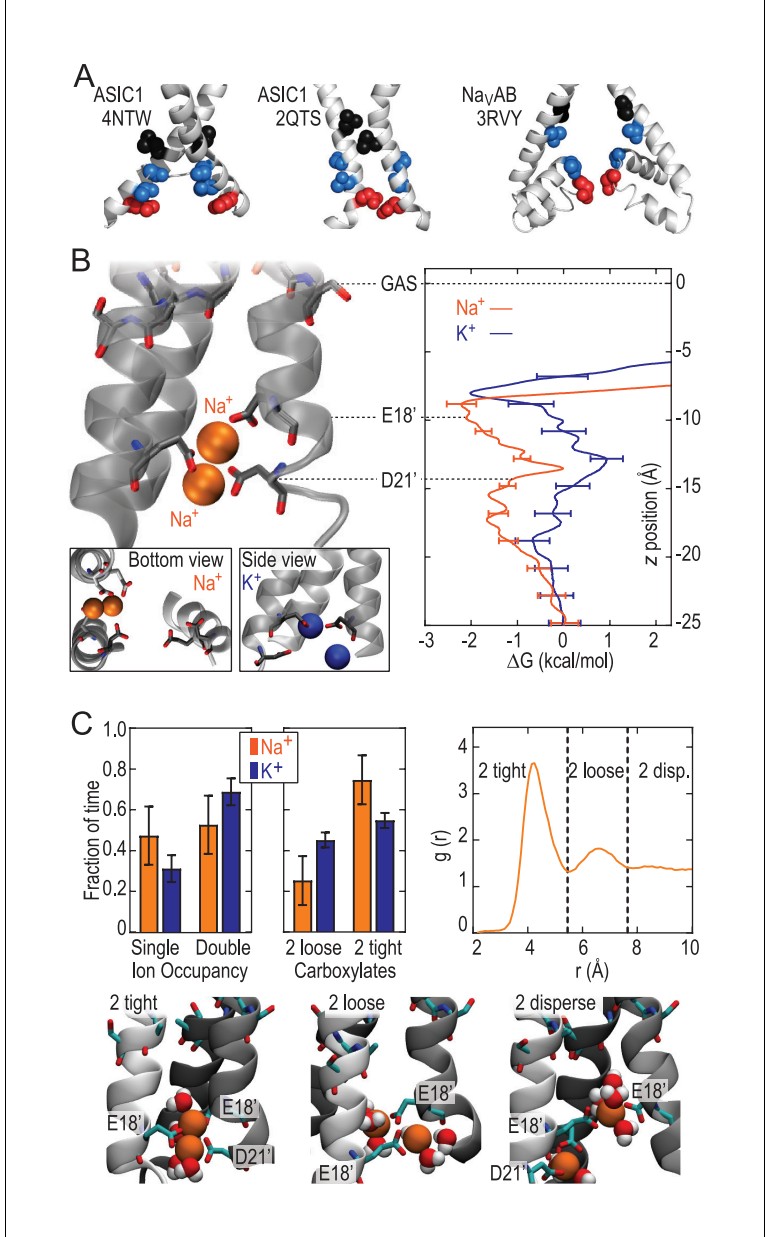

**Figure 3.** Free energy profiles from Umbrella Sampling of PDB:2QTS. (**A**) ASIC PDBs compared with voltage-gated Na⁺ channel NaᵥAb (PDB:3RVY). Two opposing helices shown, with NaᵥAb inverted for comparison. Selected leucine/isoleucine (black), serine/threonine (blue) and glutamate (red) side chains shown as spheres. (**B**) Snapshot of E18′ and D21′ creating multiple ion/carboxylate complex (left) and single ion free energy profiles (right) for Na⁺ and K⁺ in the lower pore of the presumed closed conformation PDB:2QTS (see *Figure 2—figure supplement 1* for convergence). (**C**) Distribution of single and double ion occupancy in 2QTS. Double ion occupancy dominates, usually involving multiple side chains. In this case, states are defined by the radial distribution function g(r) for ion-ion distance (right), revealing: two ions in a tightly shared complex with multiple carboxylates (2 tight); a two ion complex involving neighboring carboxylates that do not share single ions (2 loose); and the remainder involving ion binding to distant groups on opposing subunits (2 disperse). Snapshots below illustrate each configuration.

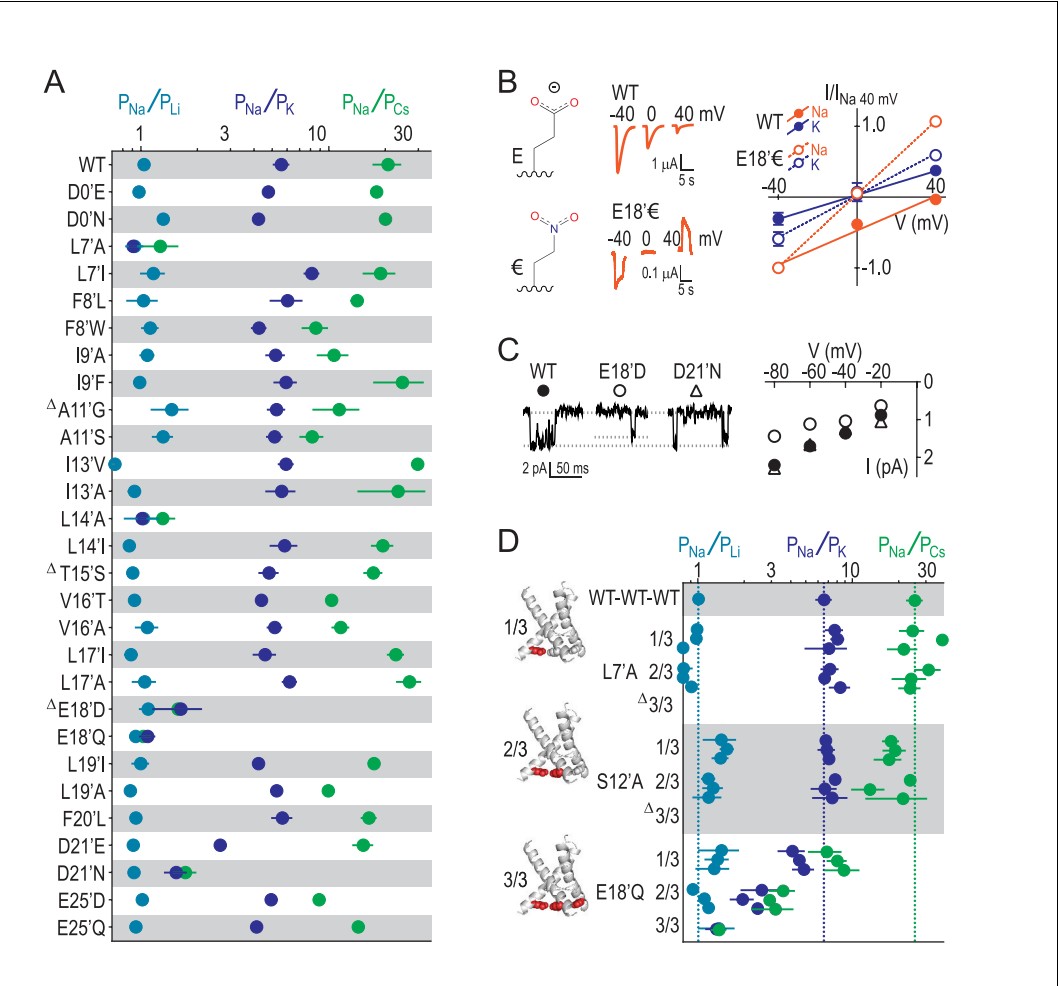

**Figure 4.** E18′ plays a direct role in selective Na$^+$ conduction. (**A**) Relative permeability ratios for WT and mutant ASIC1a channels (mean ± SEM; n = 3–6). Δ: G10′A/S, A11′V/F, S12′G/A/C/T, T15′V and E18′A channels were non-functional (see *Figure 4—figure supplement 1*). (**B**) E18′ and €18′ side chains, proton-gated currents (pH 6.0) in extracellular NaCl, and current-voltage relationship in extracellular NaCl (orange) or KCl (blue; mean ± SEM, n = 3–4). Specific € incorporation illustrated in *Figure 4—figure supplement 1*. (**C**) Example and mean (± SEM, n = 5–6) single channel Na$^+$ currents. (**D**) Relative permeability ratios for concatemeric channels carrying no mutations (WT-WT-WT) or one, two or three L7′A, S12′A or E18′Q mutations (mean ± SEM, n = 3–6). Δ: Concatemeric channels containing three L7′A- and three S12′A-mutated subunits non-functional (see *Figure 4—figure supplement 2*).

The following figure supplements are available for figure 4:

**Figure supplement 1.** Experimental identification of determinants of ion selectivity.

**Figure supplement 2.** Design and biochemical validation of concatemeric ASICs.

significantly reduced (*Figure 4C*), implying that D18′ is too short to enable WT-like Na$^+$ permeation (although single channel conductance was similar to WT in D21′N). We postulate that the need for such precise positioning of E18′ carboxylates might be a consequence of the formation of Na$^+$-selective multi-carboxylate complexes during conduction (e.g. *Figure 3C*). In particular, this suggests proximity of carboxylates from neighboring subunits, as would occur in a channel conformation with a more compact lower pore, similar to that seen in *Figure 3B*.

Finally, to establish the individual contribution of each of the three E18′ side chains lining the pore in the ASIC trimer, we generated concatemeric channels containing the E18′Q mutation in one, two or all three subunits. As shown in *Figure 4D*, each E18′ contributes to selectivity, as even the 1/

3 mutation decreases $P_{Na+}/P_{K+}$ and $P_{Na+}/P_{Cs+}$ ratios. But while 1/3 maintained WT-like profile of $P_{Na+}$ > $P_{K+}$ > $P_{Cs+}$, it was only 2/3 and 3/3 where this profile collapsed, consistent with the cooperative interactions with ions suggested by simulations (*Figure 3*). In contrast, the L7'A mutation, which abolished selectivity in regular channels, caused no ostensible change in relative permeability in 1/3- or 2/3-mutated channels (*Figure 4D*). Furthermore, the S12'A mutation rendered regular channels non-functional (*Figure 4—figure supplement 1*) and had therefore eluded study. However, a step-wise introduction of this mutation into concatemeric channels caused no change in relative permeability in 1/3- or 2/3-mutated channels, providing novel functional evidence that S12' makes little or no direct contribution to Na$^+$ selectivity.

## Discussion

These results are the first to extensively assess the role of the GAS belt with atomic resolution in ion selectivity in ASICs, as previous approaches were not suited to probing this apparently delicate part of the channel. Through experimental and computational approaches, we find that the GAS belt is involved in reducing the barrier to K$^+$ ions through backbone interactions, but we find no evidence for Na$^+$ selectivity occurring exclusively at this position. Our data identify other regions, including L7' and L14' on either side of the GAS belt, and especially E18' at the lower entrance to the pore, where selective Na$^+$ conduction is most directly controlled. This prompts a rethinking of the long-standing view that the GAS belt is *the* selectivity filter in ASICs.

### Defining the role of the GAS belt

The GXS motif was first suggested to form a filter that excludes larger ions in ENaCs, when S12'A, S12'C and S12'D mutations were shown to reduce relative Na$^+$/K$^+$ permeability (*Kellenberger et al., 1999a*; *Snyder et al., 1999*). The early notion that such a size-exclusion selectivity filter could extend to ASICs (*Kellenberger and Schild, 2002*) gained traction when X-ray structures of an apparently open-channel chick ASIC1/toxin complex showed that in kinking the pore-forming helix, GAS motifs from each subunit together form a belt in the narrowest part of the pore (*Baconguis et al., 2014*). However, nearly all attempts to probe the functional role of the GAS belt in ASICs were severely hindered by the very intimacy of this role: mutations to G10' and S12' in ASIC1a practically abolish measurable activity (*Carattino and Della Vecchia, 2012*; *Li et al., 2011b*; *Yang et al., 2009*). Finally, certain experimental data seem at odds with a size-exclusion filter in ASICs. For instance, potassium and even larger organic cations can permeate the ASIC pore, suggesting that specific interactions in the channel pore contribute to the selective conduction of Na$^+$ (*Yang and Palmer, 2014*).

One previous attempt to overcome the sensitivity of GAS to mutation succeeded by engineering a concatemer containing two WT subunits and one G10'C subunit, causing a reduction in Na$^+$ conductance (*Li et al., 2011b*). However, this mutation adds a side chain to G10', whose main chain carbonyl oxygen is implicated in conduction by structural data (*Baconguis et al., 2014*), and the G10'C mutation could indirectly alter the shape of the pore, as indeed *Kellenberger et al. (1999a)*, suggested for GAS mutations in ENaCs. We have targeted G10´ directly and yet subtly, by way of an amide-ester main chain substitution. This caused minimal structural consequences, reflected in WT-like gating of A11'α channels (*Figure 1C*), yet decreased electrostatic potential around the G10' main chain carbonyl oxygen (*Lu et al., 2001*). This alteration saw no effect on Na$^+$ conductance but a modest decrease in apparent K$^+$ permeability. Our MD simulations provide a good explanation for this effect. Na$^+$ and K$^+$ ions both fit through the GAS constriction, but for K$^+$ ions, this requires interaction with a G10' carbonyl in place of a water molecule (*Figure 2—figure supplement 2*). Additionally, we used mutant S12' concatemeric channels to directly demonstrate that S12´ is unlikely to play a major role in ASIC selectivity. Thus, our experiments and simulations show that, while the GAS belt may contribute to ion selectivity in ASICs, it is unlikely the site of the actual selectivity filter, which we have demonstrated is located elsewhere in the pore, where mutations eliminate Na$^+$ selectivity altogether.

### Determinants of ion selectivity

While the small contribution from backbone K$^+$ coordination explains the lack of ion preference at the GAS constriction, nearby site L7' caused slight dehydration of K$^+$, uncompensated for by

carbonyl coordination in a more open part of the pore, explaining its slight elevation in energy. Alanine (but not isoleucine) substitution of L14′, on the other side of the GAS belt, also affected selectivity, but with no calculated energetic difference between $Na^+$ and $K^+$. We propose that the loss of selectivity in the L14′A mutant could be an indirect effect on $Na^+$ conduction, caused by the loss of hydrophobic interactions between L14′ side chains and adjacent helices (illustrated by *Baconguis et al., 2014*), which are likely retained in the more conservative L14′I mutant. This indirect role in selectivity is consistent with the absence of an energetic preference for $Na^+$ at this level in simulations. Furthermore, a role for the L14′ side chain in maintaining the open channel structure is also consistent with the effects of the L14′C mutation on channel gating (*Carattino and Della Vecchia, 2012*). Similarly, the L7′ side chain is oriented toward adjacent helices in PDB:4NTW (*Baconguis et al., 2014*) and is implicated in gating (*Yang et al., 2009*). Indeed, in our hands the L7′A mutation altered ASIC1a proton-gating more than any other mutant in the present study (*Figure 4—figure supplement 1*). L7′ might thus be important for pore conformation, without contributing to specific interactions with ions in the pore (consistent with the hydrophobic nature of the side chain).

The most noticeable effects on ion selectivity, based on both experiments and simulations, were caused by mutations at the lower end of the channel pore, involving E18′ and D21′. In the case of E18′, our experiments showed that both charge and size of this side chain are crucial for selectivity, suggesting the need for precise charge positioning, consistent with the formation of multi-carboxylate complexes with $Na^+$ ions. Through engineered subunit stoichiometry we showed that although each E18′ contributes to selective $Na^+$ conduction (as expected with the three E18′ side chains being oriented into the pore in all available ASIC crystal structures [*Baconguis et al., 2014*; *Baconguis and Gouaux, 2012*; *Jasti et al., 2007*]), the elimination of 2 of the three glutamates drastically reduces $Na^+/K^+$ relative permeability, consistent with the notion of cooperative $Na^+$ binding to multiple glutamate side chains. We have shown that there is a significant energetic preference for $Na^+$ at the level of E18′, established by ion-carboxylate complexes that favor $Na^+$ ions, forming a uniquely ion selective site in ASIC1a. In the Appendix we demonstrate that strong binding of $Na^+$ to carboxylates in this model is also seen in high level quantum mechanical free energy calculations, yielding a $K_D$ in approximate agreement with experiments, and that preference for $Na^+$ is maintained even with more repulsive ion parameters (*Marinelli et al., 2014*). We postulate that such thermodynamic stability at the channel entrance would act to facilitate binding of multiple $Na^+$ ions to increase channel flux, possibly via knock-on permeation. As E18′ is highly conserved (*Figure 1—figure supplement 1*) and mutations at the equivalent position in ENaCs reduce $Na^+$ currents (*Langloh et al., 2000*; *Sheng et al., 2001*), it is foreseeable that E18′ controls $Na^+$ conduction throughout the broader ENaC/DEG family.

As ENaCs show substantially greater $Na^+$ selectivity than ASICs (e.g. *Gründer and Pusch, 2015*), however, significant differences must occur between these two examples of ENaC/DEG channels. We speculate that G10′ might be oriented differently in ENaCs, such that it cannot provide the favorable backbone contribution to $K^+$ conduction that we observed for ASICs. Moreover, in the selectivity filter we have identified, ENaCs possess E18′ and E21′ (whereas ASICs possess E18′ and D21′). The longer E21′ side chain in ENaCs could help in the formation of cooperative multi-ion complexes, which could further increase $Na^+$ selectivity, as suggested in our simulations with a narrower pore than that seen in PDB:4NTW. As we did not observe an increase in selectivity with D21′E mutant ASICs, we speculate that other factors govern the additional increase in selectivity in ENaCs, possibly by an even more narrow selectivity filter. This notion is consistent with greater permeation of organic cations in ASIC (*Yang and Palmer, 2014*) than in ENaC (*Kellenberger et al., 2001*). Finally, it is conceivable that the intracellular entrance to the open ASIC channel pore would adopt a different conformation in the presence of intracellular N- and C-termini, as is certainly the case in structurally analogous P2X receptors (*Hattori and Gouaux, 2012*; *Mansoor et al., 2016*).

## Concluding remarks

Our combined experimental and computational approach reveals that although main chain carbonyl oxygens at the levels of L7′ and the GAS belt contribute to selectivity by interacting with permeating $K^+$ ions, the notion of the GAS belt as the ASIC selectivity filter must be revised. Our data clearly describe a selectivity filter (comprised of E18′ and D21′) at the lower end of the ASIC1a pore, which discriminates ions based on interactions with a band of carboxylate side chains.

## Materials and methods

### Molecular biology

Mouse ASIC1a cDNA, cloned between BamHI and SacI sites of pSP64 vector, was a gift from Marcelo Carattino (University of Pittsburgh). In screening for determinants of ion selectivity (*Figure 4A*), two single amino acid substitutions per position were designed to identify loss-of-function without relying on results from only one substitution. Site-directed mutagenesis used custom-designed primers (Eurofins Genomics, Germany) and regular PCR with PfuUltra II Fusion HS DNA Polymerase (Agilent Technologies, Denmark). For generation of concatemeric constructs (illustrated in *Figure 4—figure supplement 2*), the ASIC1a insert was cloned out of pSP64 with forward and reverse primers containing additional HindIII and SalI, SalI and BamHI or BamHI and SacI sequence, respectively, generating three distinct inserts, *a*, *b* and *c*, which were then gel-purified. Insert *a* was ligated into pSP64 vector (Promega, Denmark) double-digested with HindIII and SalI. The resulting plasmid was then then double-digested with SalI and BamHI, insert *b* was ligated in; this '*a-b*' plasmid was double-digested with BamHI and SacI, and insert *c* was ligated in, yielding the '*a-b-c*' concatemer. From this construct, wild-type inserts could be replaced with mutant inserts via the same primers and restriction sites. Full concatemer sequences were confirmed by mutagenesis primers recognizing the unique restriction sites. cDNAs were linearized with EcoRI, and mRNAs were synthesized with the Ambion mMESSAGE mMACHINE SP6 transcription kit (Thermo Fisher Scientific, Denmark). mRNA was then purified via RNeasy columns (Qiagen, Denmark).

### Incorporation of unnatural amino acids, oocyte preparation and mRNA injection, and Western Blotting

The non-proteinogenic α-amino and α-hydroxy acids, or 'unnatural amino acids' (UAAs), 4-nitro-2-aminobutyric acid (€) and lactic acid (hydroxy analog of alanine, α) were incorporated via the nonsense suppression method, detailed elsewhere (*Dougherty and Van Arnam, 2014*). Briefly, modified *Tetrahymena thermophila* tRNA (*Nowak et al., 1998*) was prepared by ligating full-length 5' and 3' DNA strands (Integrated DNA Technologies, Belgium), RNA synthesis with T7-Scribe transcription kit (Cellscript, WI, USA) and purification with Chroma Spin DEPC-$H_2O$ columns (Clontech, CA, USA). UAAs were ligated to tRNA with T4 DNA ligase (New England Biolabs, MA, USA), and aminoacyl-tRNA was purified with phenol-chloroform extraction and ethanol precipitation. € was synthesized as described elsewhere, (*Cashin et al., 2007*) protected with 4,5-dimethoxy-2-nitrobenzyl carbamate (NVOC), esterified with 5'-O-phosphoryl-2'-deoxycytidylyl-(3'–>5')adenosine (pdCpA), ligated to tRNA, purified via ethanol precipitation, air-dried and stored at −80°C until use. Immediately before injection into oocytes, €-tRNA was resuspended in 1 μl water and deprotected with UV light. For α, the lactic acid was esterified with pdCpA as described elsewhere (*England et al., 1999*), the resulting ester was ligated to tRNA and purified as above, but required no de-protection before injection.

Ovaries surgically removed from *Xenopus laevis* frogs (anaesthetized in 0.3% tricaine, under license 2014–15−0201−00031, approved by the Danish Veterinary and Food Administration) were divided into small clumps of oocytes, these were shaken at 200 rpm at 37°C in OR2 (in mM, 2.5 NaCl, 2 KCl, 1 $MgCl_2$, 5 HEPES, pH 7.4 with NaOH) containing 0.5 mg/ml Type I collagenase (Worthington, NJ, USA) until free individual oocytes were obtained. These were incubated in OR2 at 18°C until injection of mRNA. For WT and most mutant mRNAs, 0.4 ng was injected in a volume of 36 nl (diluted in water) using a Nanoliter 2010 injector (World Precision Instruments, UK). For mutant and concatemeric constructs that yielded lower maximum currents in electrophysiological recordings, 36 ng (in 36 nl) were injected. For UAA incorporation, 3.6 ng (for A11'α) 36 ng (for all others) UAG-mutant mRNA together with half a pellet of UAA-tRNA ligate in a volume of 46 nl was injected. Oocytes were incubated in Leibovitz's L-15 medium (Gibco, Denmark) with 3 mM L-glutamine, 2.5 mg/ml gentamycin, 15 mM HEPES (pH 7.6 with NaOH) until experiments.

For Western blotting, cell surface protein was purified with Pierce Cell Surface Protein Isolation Kit (this and subsequent kits/reagents from ThermoFisher Scientific), modified here for oocytes: 30 oocytes were injected with a single construct and used for cell surface protein labeling and purification as per supplier's instructions. Protein (50 μg per construct) was denatured, separated in a 4–12% BIS-TRIS gel and transferred to PVDF membrane, as per supplier's instructions. Membrane was

washed with rabbit polyclonal anti-ASIC1a antibody (OSR00097W), washed with goat anti-rabbit IgG/HRP conjugate (A16110), and developed with Pierce ECL (32106).

## Electrophysiological experiments and data analysis

For two-electrode voltage clamp experiments, one to two days after injection, an oocyte was placed in a custom-built chamber (*Dahan et al., 2004*), through which bath solution (in mM, 96 NaCl, 2 KCl, 1.8 $BaCl_2$, 5 HEPES, pH 7.6 with NaOH) was continuously perfused. Solution was rapidly switched to one of lower pH (for those <6.0, HEPES was replaced with MES) using a ValveBank eight system (AutoMate Scientific, CA, USA). Oocytes were clamped at −40 mV (or as described in text) and currents recorded with microelectrodes filled with 3 M KCl, OC-725C amplifier (Warner, CT, USA) and Digidata 1550 digitizer (Molecular Devices, CA, USA) at 1 kHz with 200 Hz filtering. Data were later analyzed in Clampfit 10 (Molecular Devices) with 10 Hz filtering for illustration. In measuring ion selectivity, NaCl was replaced with LiCl, KCl or CsCl and pH was adjusted with NaOH and HCl (LiCl solutions) or CsOH and HCl (KCl and CsCl solutions). For single channel recordings, oocytes were prepared in the same way until recording, when the vitelline membrane was removed with forceps, and outside-out patches were pulled. Intracellular solution contained, in mM, 20 NaCl, 180 D-Mannitol, 2 $MgCl_2$, 5 HEPES, pH 7.4 with NaOH/HCl; extracellular solution 140 NaCl, 3 $MgCl_2$, 5 HEPES, pH 7.6 or 5.0 with NaOH/HCl). Currents were recorded with an Axopatch 200B amplifier and Digidata 1550A digitizer (Molecular Devices) at 10 kHz with 2 kHz filtering. Data were analyzed in Clampfit 10 with additional 500 Hz filtering for analysis and display.

Relative ion permeability (e.g. $P_{Na}/P_K$) was calculated using reversal potentials ($V_{rev,Na}$ and $V_{rev,K}$) measured from 160 mV/200 ms ramps and a version of the Goldman-Hodgkin-Katz equation $P_K/P_{Na} = \exp(F(V_{rev,Na} - V_{rev,K})/RT)$, where F = Faraday's constant, R = gas constant, T = 274 K, as performed elsewhere, where it was shown that voltage ramps yield results indistinguishable from voltage intervals or 'steps' (*Yang and Palmer, 2014*). Relative ion permeability ratios are plotted as $P_{Na}/P_{Ka}$ on a log scale for clarity. $pH_{50}$ values were calculated with the four parameter Hill equation in Prism 6 (GraphPad). As per the norm in electrophysiological experiments, where n is usually <10, we show a few individual, raw recordings as examples and calculated the mean from 3 to 9 experiments. Statistical comparisons are as described in text, but generally, multiple values were compared with one-way ANOVA and Dunnett's test for significant difference from wild-type. All functional experiments were performed on at least two batches of cells (isolated from two different animals).

## System construction and molecular dynamics simulation

For molecular dynamics (MD) simulations, ASIC1a protein X-ray crystallographic coordinates for the presumed open state, taken from PDB:4NTW (*Baconguis et al., 2014*) (available residues 45 to 456; pdb numbering) was embedded in a lipid bilayer of 386 (194 top and 192 bottom) palmitoyloleoyl-phosphatidylcholine (POPC) lipids, with explicit water molecules (47,184 molecules), 150 mM of NaCl or KCl (162 $Na^+$ or $K^+$ and 123 $Cl^-$ ions), to form two simulation boxes of 115 x 115 × 160 Å containing 213,043 atoms (*Figure 2A*).

In independent simulations of the presumed closed state, X-ray crystallographic coordinates from PDB:2QTS (*Jasti et al., 2007*) (available residues 42 to 458, 42 to 461, and 40 to 457 for each subunit, respectively; PDB numbering,) was embedded in a bilayer of 412 (206 top and 206 bottom) dipalmitoylphosphatidylcholine (DPPC) lipids, with explicit water molecules (49,850 molecules), 150 mM of NaCl or KCl (188 $Na^+$ or $K^+$ and 136 $Cl^-$ ions), to form two simulation boxes of 115x115 × 164 Å containing 223,315 atoms.

All systems were built and equilibrated with the CHARMM program (*Brooks et al., 1983*, *2009*), using the C36 lipid (*Klauda et al., 2010*) and C22 protein parameters (*MacKerell et al., 1998*) with CMAP corrections (*Mackerell et al., 2004*), and TIP3P water (*Jorgensen et al., 1983*). Ion parameters used are those for the CHARMM27 force field with revisions to $Na^+$ and $K^+$ non-bonded Lennard-Jones (*Noskov and Roux, 2008*) (referred to as CHARMM27*), corresponding to $r_{min}$ values for ion-carboxylate oxygen pair interactions of 3.12 and 3.46 Å, respectively. In the Appendix we demonstrate that these parameters yield reasonable agreement with experimental binding for $Na^+$ in aqueous solution and to quantum mechanical calculations. We also report modified results using NBFIX parameters for $Na^+$ and $K^+$ that better reproduce experimental osmotic pressure coefficients in concentrated acetate salt solutions (*Marinelli et al., 2014*), corresponding to

$r_{min}$ values of 3.19 and 3.52 Å, respectively (see Appendix). Modified Lennard-Jones parameters were used to describe the interactions between cations and carbonyl oxygen atoms of the protein, to reproduce correct free energies of solvation in protein backbone mimetic, N-methylacetamide (*Allen et al., 2006*; *Bernèche and Roux, 2001*; *Noskov et al., 2004*). Standard CHARMM LJ parameters for Cl⁻ were used (*Beglov and Roux, 1994*). After 3000 (2 x 6 × 250) steps of steepest descent (SD) and adopted basis Newton-Raphson (ABNR) minimization, MD simulations commenced with initial harmonic restraints (10 kcal/mol/Å$^2$) applied to all heavy atoms. These restraints were slowly released over 0.5 ns, followed by 1.5 ns of simulation without any restraints. Simulations were started at constant volume with a timestep of 1 fs for 50 ps before switching to constant pressure (one atm, Langevin piston barostat) (*Andersen, 1980*; *Feller et al., 1995*) with a timestep of 2 fs, and constant temperature using a Nosé-Hoover thermostat (*Hoover, 1985*; *Nosé, 1984*). All simulations of POPC were run at 303K (above the gel phase transition temperature of 271K), and those for DPPC at 323K (above the transition temperature of 314K) (*Davis, 1979*; *Petersen et al., 1975*). All bonds to H atoms were maintained using the SHAKE algorithm (*Ryckaert et al., 1977*). Electrostatic interactions were computed using Particle Mesh Ewald (*Darden et al., 1993*), with grid spacing of 1 Å and 6$^{th}$ order B-spline for mesh interpolation. Non-bonded pair lists were updated automatically via heuristic testing with a cutoff distance of 16 Å and a real space cutoff of 12 Å with energy switch (switching distance of 10 Å).

Production simulations used the NAMD program, version 2.9 (*Phillips et al., 2005*) with the same force field. These simulations were carried out using tetragonal periodic boundary conditions in the NPT ensemble with a two fs time step. All simulations were performed at constant pressure (1 atm) with a Langevin piston, (*Andersen, 1980*; *Feller et al., 1995*) with fixed lateral area ratio (1:1), and constant temperature using a Nosé-Hoover thermostat (*Hoover, 1985*; *Nosé, 1984*), All bonds to H atoms were maintained using the RATTLE algorithm (*Andersen, 1983*). Electrostatic interactions were computed using Particle Mesh Ewald with grid spacing of 1.5 Å and 6$^{th}$ order B-spline for mesh interpolation. Non-bonded pair lists were updated every 20 fs with a cutoff distance of 16 Å and a real space cutoff of 12 Å with energy switch (switching distance of 10 Å).

Unbiased simulations of PDB:4NTW ran for 235 and 225 ns in NaCl and KCl, respectively. To ensure sampling of the lower channel entrance in the non-conducting PDB:2QTS state, flat bottom constraints were applied to two ions to restrain them within a cylinder of 15 Å of radius along the z-axis, and in a slab of 10 Å of thickness in the xy-plane, both centered around the center of mass (COM) of the backbone of residues E18' and D21'. Simulations in 2QTS ran for 140 ns in both NaCl and KCl solutions.

## Free energy profiles

The unbiased free energy profile for PDB:4NTW (*Figure 2—figure supplement 1*) was calculated as $W(z) = -k_B T \ \text{in}[\rho(z)] + C$, where ρ is the unbiased probability distribution as a function of reaction coordinate z, being the position of the ions along the z-axis, with constant C chosen to set the zero in bulk electrolyte. The mean and standard error were obtained by dividing data into four equal blocks of 56 ns for each simulation.

To ensure thorough sampling of free energies for a single Na⁺ or K⁺ ion translocating the 4NTW pore, or for entering the lower pore of 2QTS, we used umbrella sampling (US) (*Torrie and Valleau, 1977*). This involved 46 and 26 independent simulations (windows) for 4NTW and 2QTS, respectively, with 1 Å resolution: windows spanning −25 to 20 Å in 4NTW, and −25 to 0 Å in 2QTS, relative to the GAS sequence COM. Initial configurations were taken from the unbiased simulations of 4NTW and 2QTS in NaCl and KCl after 20 and 12 ns of equilibration, respectively. In each window, the ion was held near the window position by a 2.5 kcal/mol/Å$^2$ force constant. To assist sampling of K⁺ in 2QTS, two additional windows were added z = 9.5 Å and z = 11.5 Å, and the force constant was increased to 5.14 kcal/mol/Å$^2$ for Na⁺ at z = 7 Å. The radial position was constrained with a flat bottom potential to keep the ion in a cylinder of 9 Å with a force constant 10 kcal/mol/Å$^2$. All US simulations were performed with NAMD2.9 at 1 atm constant pressure using the Langevin piston barostat (*Feller et al., 1995*), and constant temperature of 323K using a Nosé-Hoover thermostat (*Hoover, 1985*; *Nosé, 1984*). We ran 20 ns per window (extended to 27 ns for the central windows in 4NTW, z = −10 to z = −5 Å). Based on convergence for each US window (*Figure 2—figure supplement 1*), we discarded the first 6 ns for Na⁺ and 7 ns for K⁺ simulations in 2QTS, and the first 6 ns for Na⁺ and 3 ns for K⁺ simulations in 4NTW. Unbiased free energy profiles were then calculated

using the Weighted Histogram Analysis Method (WHAM) (*Kumar et al., 1992*). Mean and standard error were calculated by dividing the data into 1 ns blocks.

## Free energy perturbation

Free energy perturbation (FEP) calculations (*Kollman, 1993*) were performed for ions in a site formed by the carboxyl oxygen atoms of residues E18' and D21', involving 2 to 4 residues from different subunits (systems 4NTW and 2QTS), as well as in the site formed by the carbonyl oxygen atoms of the GAS locus (system 4NTW only). At the level of residues E18' and D21', single or double occupancies ($Na^+$ or $K^+$), identified from analysis of the unbiased simulations above, were maintained with a flat-bottom spherical constraint of 2.75 Å radius from the COM of the carboxyl groups forming the site. In the GAS site, flat-bottom planar potentials were applied to hold the ion in a slab of 4 Å of thickness, oriented in the XY-plane and centered on the COM of the GAS backbone.

The relative change in free energy resulting associated with the binding of $Na^+$ or $K^+$ to a given site and occupancy can be summarized by the following thermodynamic cycle:

$$\begin{array}{ccc} Na^+_{bulk} & \rightarrow & K^+_{bulk} \\ \uparrow & & \downarrow \\ Na^+_{site} & \rightarrow & K^+_{site} \end{array}$$

Accordingly, this implies that the resulting variation of free energy $\triangle\triangle G$ can be determined as:

$$\Delta\Delta G(Na^+ \rightarrow K^+) = [(G_{site}(K^+) - G_{bulk}(K^+)) - (G_{site}(K^+) - G_{bulk}(K^+))],$$

or:

$$\Delta\Delta G(Na^+ \rightarrow K^+) = \Delta G_{site}(Na^+ \rightarrow K^+) - G_{bulk}(Na^+ \rightarrow K^+)$$

The free energy differences $\Delta G_{site}(Na^+ \rightarrow K^+)$ and $\Delta G_{bulk}(Na^+ \rightarrow K^+)$ were estimated using alchemichal FEP simulations, in which the thermodynamic work for the alchemical transformation of the ion was calculated step-by-step, slowly mutating the ion from one species to the other using a coupling parameter λ. These calculations were performed with the alchemical FEP module implemented in NAMD 2.9. For each calculation, λ was initially 'pulled' from λ = 0 to 1 (forward simulation) and 1 to 0 (backward simulation) in 40 steps (Δλ = 0.05) of 0.2 ns each, followed by 2 ns of production run in parallel in each window, amounting to 80 ns of simulations per calculation, offering good convergence (see *Figure 2—figure supplement 1*). All simulations were performed with the same simulation parameters, as described for free energy profiles above. The mean and standard error for each calculation estimated by combining forward and backward calculations using the Bennett acceptance ratio (BAR) (*Bennett, 1976*) estimator implemented in the *ParseFEP* plugin of the VMD package (*Humphrey et al., 1996*).

A total of 20 FEP calculations were performed in the sites formed by residues 18' and 21' (4NTW and 2QTS), two in the GAS site (4NTW) and two in the bulk to provide a reference for binding. For both systems (2QTS and 4NTW), at the level of residues E18' and D21', we considered single occupancy ($Na^+\rightarrow K^+$ and $K^+\rightarrow Na^+$) and double occupancy, where each ion was transformed in a separate calculation ($Na^+Na^+\rightarrow Na^+K^+$ and $Na^+Na^+\rightarrow K^+Na^+$, as well as $K^+K^+\rightarrow K^+Na^+$ and $K^+K^+\rightarrow Na^+K^+$), followed by the transformation of the second ion, once the first one reached the λ = 1 state ($Na^+K^+\rightarrow K^+K^+$ and $K^+Na^+\rightarrow K^+K^+$, as well as $K^+Na^+\rightarrow Na^+Na^+$ and $Na^+K^+\rightarrow Na^+Na^+$). Free energy changes for the double occupancy transformation $Na^+Na^+\rightarrow K^+$ were obtained by summing the changes along the path, $Na^+Na^+\rightarrow K^+Na^+$ and $Na^+K^+\rightarrow K^+K^+$. Results are summarized in *Figure 2—figure supplement 1*.

## Acknowledgements

We thank Hiu Li for assistance in computing CPMD free energy profiles, and Igor Vorobyov for important discussions on ion parameter testing. The authors would like to thank Dr. Harley Kurata for comments on the manuscript. We acknowledge the Lundbeck Foundation (R171-2014-558; TPL, R139-2012-12390; SAP), Danish Council for Independent research (4092-00348B; TPL), Carlsberg Foundation (2013_01_0439; SAP), NHMRC (APP1104259; TWA), NIH (U01-11567710, TWA), ARC (DP170101732; TWA), Medical Advances Without Animals Trust and NCI (dd7; TWA) for support.

## Additional information

### Funding

| Funder | Grant reference number | Author |
|---|---|---|
| Det Frie Forskningsråd | Postdoctoral Fellowship 4092-00348B | Timothy Lynagh |
| Lundbeckfonden | Postdoctoral Fellowship R171-2014-558 | Timothy Lynagh |
| Australian Research Council | Project Grant DP170101732 | Toby W Allen |
| National Health and Medical Research Council | Project Grant APP1104259 | Toby W Allen |
| National Institutes of Health | Project Grant U01-11567710 | Toby W Allen |
| National Computational Infrastructure | dd7 | Toby W Allen |
| Lundbeckfonden | Lundbeck Foundation Fellowship R139-2012-12390 | Stephan A Pless |
| Carlsbergfondet | Equipment Grant 2013_01_0439 | Stephan A Pless |
| Novo Nordisk Foundation | Project Grant | Stephan A Pless |

The funders had no role in study design, data collection and interpretation, or the decision to submit the work for publication.

### Author contributions

TL, Conceptualization, Data curation, Formal analysis, Supervision, Funding acquisition, Visualization, Writing—original draft, Writing—review and editing; EF, CB, Data curation, Formal analysis, Visualization, Writing—review and editing; MW, Data curation, Formal analysis; VVK, Resources, Writing—review and editing; JMC, Resources, Methodology; TWA, Conceptualization, Supervision, Formal analysis, Funding acquisition, Visualization, Writing—original draft, Project administration, Writing—review and editing; SAP, Conceptualization, Supervision, Funding acquisition, Project administration, Writing—review and editing

### Author ORCIDs

Timothy Lynagh, http://orcid.org/0000-0003-4888-4098
Stephan A Pless, http://orcid.org/0000-0001-6654-114X

### Ethics

Animal experimentation: This study was performed in accordance with the recommendations by the by the Danish Veterinary and Food Administration and approved under license 2014−15−0201−00031. Surgery was performed on Xenopus laevis frogs anaesthetized in 0.3% tricaine.

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

## Appendix 1

### Sodium-carboxylate interaction tests

Although the environment in the ion channel differs from aqueous solution, it is important to evaluate the strength of ion-carboxylate interactions in a water environment and relate to experiments and quantum mechanical (QM) calculations. *Appendix 1—figure 1* reports the results of simulations for the binding of $Na^+$ to an acetate molecule in water. We carried out Umbrella Sampling simulations for the association of these ions in a periodic water box. A range of systems with 75–500 water molecules were compared, yielding similar results to within 1 kcal/mol, leading to the choice of the smaller 75 water system to facilitate Car-Parrinello MD (*Car and Parrinello, 1985*) simulations (CPMD; performed by Hui Li, U Chicago) to compare classical results.

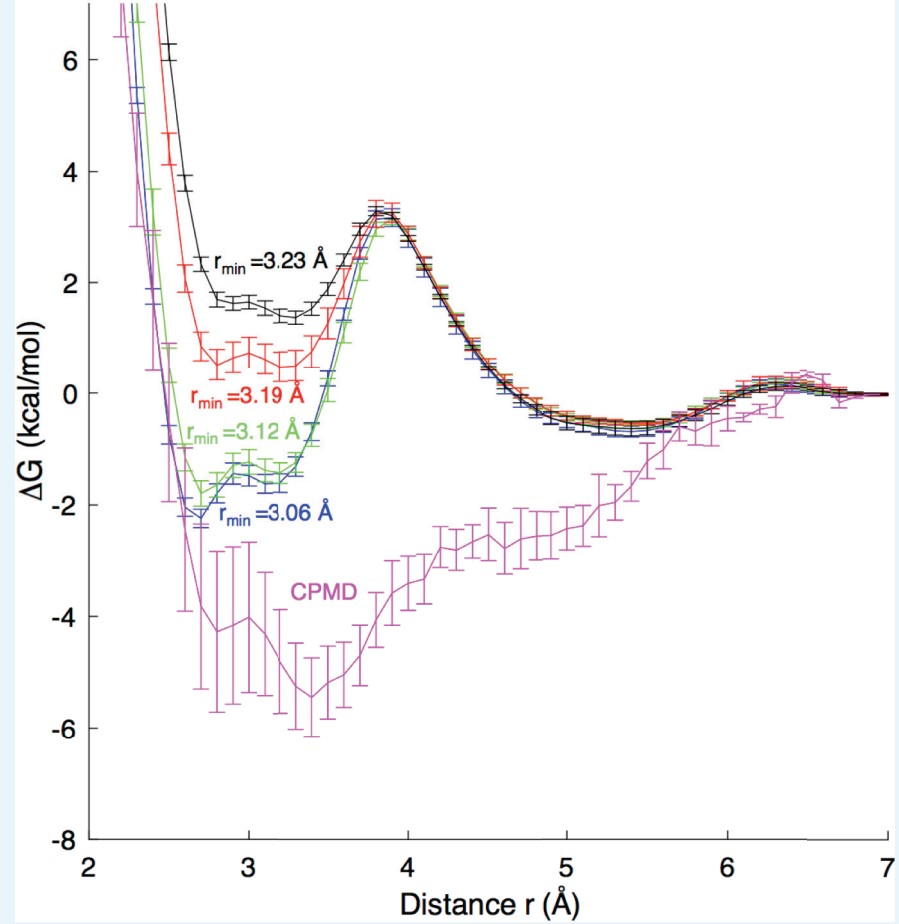

**Appendix 1—figure 1.** Free energy profiles for $Na^+$ - carboxylate binding with different pair parameters, compared to CPMD quantum mechanical calculations. Estimated dissociation constants ($K_D$) for each MD model are: CHARMM27 ($r_{min}$ = 3.06 Å) 0.6 ± 0.2 M; CHARMM27* ($r_{min}$ = 3.12 Å) 0.9 ± 0.2 M, CHARMM27* NBFIX ($r_{min}$ = 3.19 Å) 1.8 ± 0.2 M; and CHARMM27* NBFIX ($r_{min}$ = 3.23 Å) 1.8 ± 0.3 M; to be compared with experimental values discussed in the Appendix.

For classical MD simulations we carried out simulations in the NPT ensemble, with details as in the main simulations reported above, but with temperature 300K. A total of 51 Umbrella

Sampling windows (in 0.1 Å steps) were simulated for 20 ns each, with the distance between the ion and acetate center of mass constrained with a harmonic potential with force constant 230 kcal/mol/Å $^2$. These simulations were sufficient to sample pair rotation, thus incorporating the Jacobian for spherical coordinates (see below), with convergence to within 0.2 kcal/mol achieved. We have compared different force field terms for the interaction of $Na^+$ with carboxylate oxygen atoms, including: the revised Charmm27* using mixing rules to yield $r_{min}$ = 3.12 Å (**Noskov and Roux, 2008**) the standard Charmm27 mixing rules to yield $r_{min}$ = 3.06 Å; the modified LJ NBFIX used to fit to osmotic pressure for 3.0M sodium acetate solution by **Marinelli et al. (2014)** $r_{min}$~3.19 Å; and the recent $r_{min}$ of 3.23 Å from (**Venable et al., 2013**). To enable comparisons with **Marinelli et al. (2014)**, we have calculated errors for these free energies as standard deviations from 5 blocks of simulation. $K_D$ values, reported in the **Appendix 1—figure 1** caption were calculated using the same expression as **Marinelli et al. (2014)**. We note that this expression includes the Jacobian term for spherical coordinates, and as such, required that the MD profiles in **Appendix 1— figure 1** be corrected by adding $2k_BTlnr$.

For the QM profile, the Car-Parrinello MD (CPMD) 3.15 package (**Marx and Hutter, 2009**) was used, employing the BLYP exchange correlation functional with Kleinman-Bylander pseudopotential (**Kleinman and Bylander, 1982**). We used a cubic cell of dimensions 13.2 × 13.2×13.2 Å, temperature 300 K, energy cutoff 80.0 Ry, timestep 0.1 fs, and electron friction mass 400 a.u. The distance between the acetate center of mass and the cation was restrained with a harmonic force constant of 627.51 kcal/mol Å$^2$, with 50 independent simulations (windows) 0.53 Å apart, each run for 20 ps and converged to within 0.4 kcal/mol. Errors in the CPMD results are reported as ± half the deviation between profiles for first and second halves of the simulation, set to zero at r = 7 Å.

While no model is in good agreement with the CPMD profile, which itself may contain errors due to the level of QM and extent of configurational sampling (an inherent limitation in *ab initio* MD simulations), the results suggest that there is strong binding of $Na^+$ to carboxylate, which is best achieved with the unmodified ion parameter in CHARMM27*, whereas the parameter used to reproduce osmotic pressure (NBFIX $r_{min}$=3.19 Å, **Marinelli et al., 2014**), appears too repulsive. $K_D$ estimates can be compared to different available experimental values of 1.17–1.51 M (**Robinson and Stokes, 2002**) and 1.0–1.6 M (**Fournier et al., 1998**) using potentiometry in dilute sodium acetate solutions at 298K (although $K_D$ values below 1M are seen at elevated temperatures [**Fournier et al., 1998**]). The value of 0.9 ± 0.2 M for standard CHARMM27* is at the lower end of the range, while the NBFIX $r_{min}$ = 3.19 Å value of 1.8 ± 0.2 M is at the top end of the range. In further support of a lower $K_D$, we refer to separate non-polarizable and polarizable force field models using another force field (Amber), yielding negative contact free energy and equilibrium constants that correspond to $K_D$ of 0.70–0.85 M (**Iskrenova-Tchoukova et al., 2010**; **Annapureddy and Dang, 2012**).

To estimate the effects of changing to more repulsive ion-carboxylate parameters, we have computed modified analysis in **Appendix 1—figures 2** and **3**, corresponding to a change from CHARMM27* to the repulsive NBFIX values (3.19 and 3.52 Å for $Na^+$ and $K^+$, respectively). The estimated perturbation free energy for each ion position in Umbrella Sampling, from the average of Boltzmann factor to the LJ perturbation energy over the entire unperturbed trajectory, was added to the unperturbed free energy profile to yield the results in **Appendix 1—figure 2A and B**, with error bars standard errors of mean from one ns blocks of data, combined with the errors in the unperturbed results. We note the inherent approximate nature of these post-simulation estimates, as they represent the cost of overlapping ion and carboxylate oxygens as the LJ $r_{min}$ is increased, without relaxing the configurations. The result of this change of parameter is to add up to ~0.4 kcal/mol in 4NTW (**Appendix 1—figure 2A**, to compare to the original **Figure 2B**), but maintaining a similar small preference for $Na^+$ in the vicinity of E18' and D21', consistent with our experiments, and our expectation that carboxylates favor the binding of $Na^+$ (e.g. **Roux, 2010**). In the case of 2QTS (**Appendix 1—figure 2B**, to compare to the original **Figure 3B**), the more repulsive ion-carbonyl potentials lead to unfavorable energies deep inside the closed pore,

as expected, but with a binding at the lower entrance near D21' (−20 to −15 Å), favoring Na⁺. Features near z=-8 Å, deep inside the channel, are considered unreliable, with increased error and influence of the closed 2QTS pore.

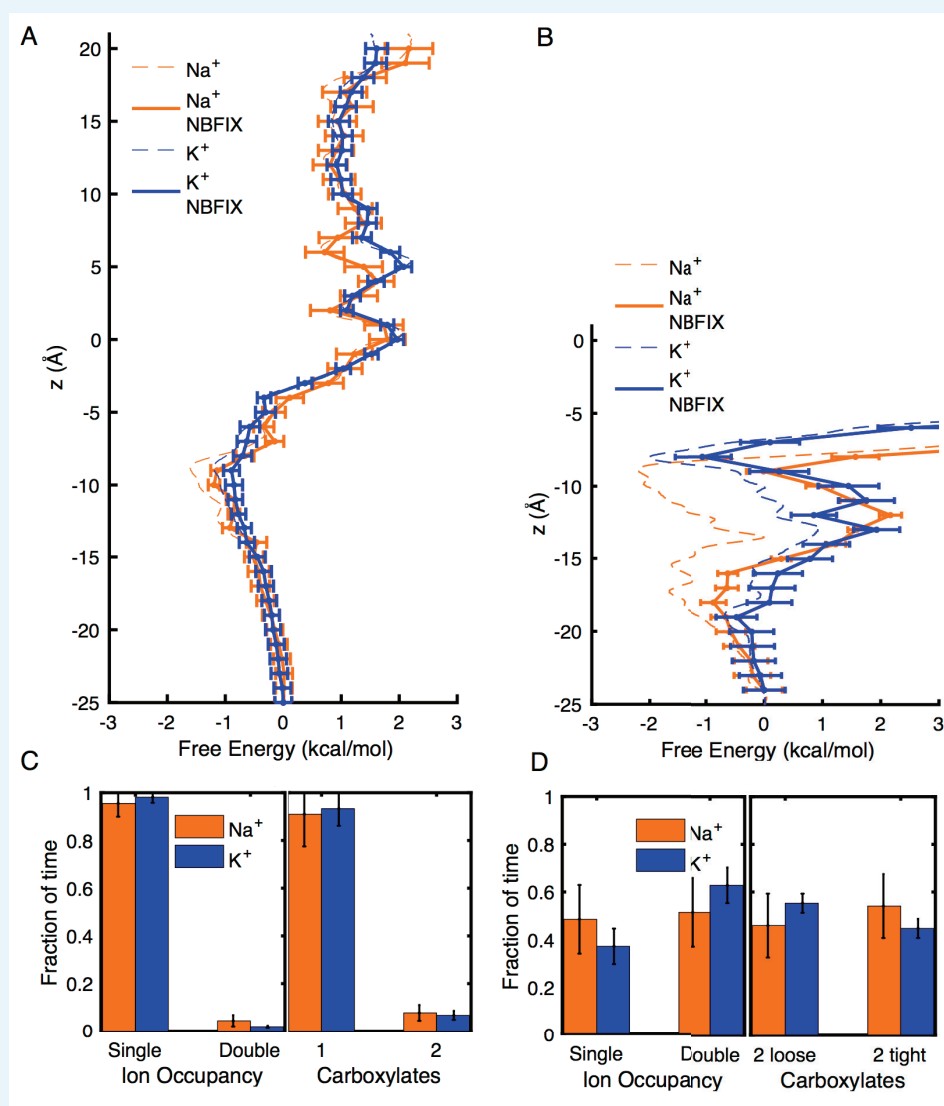

**Appendix 1—figure 2.** Estimates of free energy and coordination results for modified CHARMM27* NBFIX (Na⁺-carboxylate $r_{min}$ = 3.19 Å and K⁺-carboxylate $r_{min}$ = 3.52 Å), using post-simulation adjustment for the LJ parameter (see Appendix text). (**A**) Umbrella Sampling for Na⁺ and K⁺ crossing the pore of 4NTW. A preference of ~0.5 kcal/mol for Na⁺ over K⁺ remains around E18' and D21' (**B**) Umbrella Sampling for Na⁺ and K⁺ crossing the pore of 2QTS. Na⁺ binding and preference of ~1 kcal/mol can be observed around D21'. Values deep in the channel (above −10 Å) are considered unreliable in this closed pore. (**C**) Adjusted analysis of ion occupancies for 4NTW. Double ion occupancies are reduced by these more repulsive ion parameters, but remain more common for Na⁺ than for K⁺, (**D**) Adjusted analysis of ion occupancies for 2QTS. Double ion occupancy is favoured both for Na⁺ and K⁺, but Na⁺ ions prefer tight clusters, as seen with the unmodified parameter.

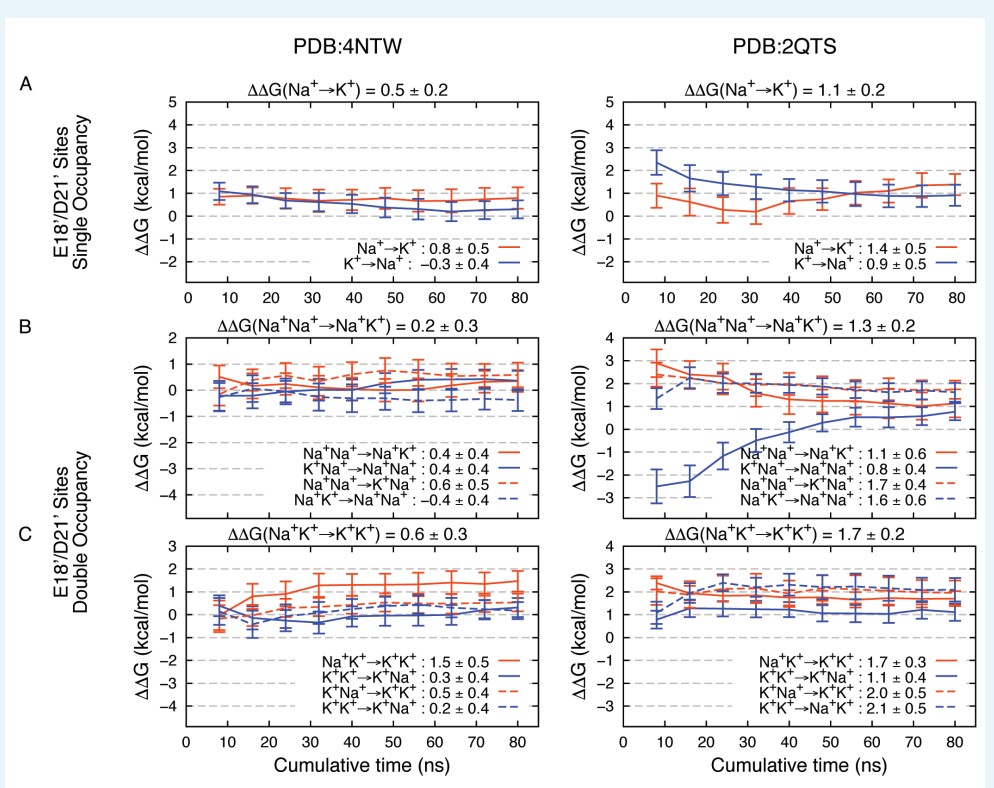

**Appendix 1—figure 3.** Estimated Na⁺→K⁺free energy perturbations for PDBs 4NTW (left graphs) and 2QTS (right graphs) with modified ion parameters. Results in panels A, B and C are as in *Figure 2—figure supplement 1B,C and D*, respectively, but with change in ion-carboxylate parameters to that of *Marinelli et al. (2014)*, with $r_{min}$ = 3.19 Å for Na⁺ and $r_{min}$ = 3.52 Å for K⁺, instead of the usual mixing rules for CHARMM27*.

Analysis of probabilities for forming different complexes in 4NTW and 2QTS channels were also modified using this change of parameter, by re-weighting each frame of the trajectory using the Boltzmann factor to the LJ perturbation energy. The reweighted analysis in *Appendix 1—figure 2C and D* (to compare to *Figures 2C* and *3C*), shows some reduction in double occupancy and tight binding of pairs, as expected. However, the probability for tight ion pairs remains higher for Na⁺ than K⁺.

Modifications to FEP results (*Appendix 1—figure 3*) tell us that these complexes remain robustly in favor of Na⁺, reinforcing the important role for E18'/D21' in Na⁺ selectivity. The adjustments reveal maintenance of 4NTW free energy difference of 0.2–0.5 kcal/mol for Na⁺→K⁺, and +0.8 kcal/mol for the Na⁺Na⁺→ K⁺K⁺ double transformation free energy (sum of Na⁺Na⁺→ Na⁺K⁺ and Na⁺K⁺→ K⁺K⁺ free energies). In 2QTS we see favorable complexes for Na⁺ by 1.1 kcal/mol for the single ion, and 1.3 kcal/mol for the 2-ion complex, with a net dual ion transformation free energy (Na⁺Na⁺→ K⁺K⁺) of 3.0 kcal/mol, strongly favoring Na⁺-carboxylate complexes, regardless of the choice of parameter.

