## [Decision Letter]

Thank you for submitting your article "A selectivity filter at the intracellular end of the acid-sensing ion channel pore" for consideration by *eLife*. Your article has been favorably evaluated by Richard Aldrich (Senior Editor) and three reviewers, one of whom, Baron Chanda (Reviewer #3), is a member of our Board of Reviewing Editors.

The reviewers have discussed the reviews with one another and the Reviewing Editor has drafted this decision to help you prepare a revised submission.

Summary:

Lynagh et al. seek to explain the structural basis for the Na^+^ versus K^+^ selectivity of ASIC channels, by combining electrophysiological, unnatural amino acid mutagenesis with molecular simulations of ion selectivity. The outcome is an innovative and compelling study, which concludes that the 10-fold Na^+^ selectivity of this channel is not determined by the so-called 'G-A-S belt', as previously proposed. Instead, a set of acidic residues at the intracellular side of the pore, and regions near Leu7/Leu14, are found to be influential. The study illustrates that even though from a structural standpoint, the ions are likely to interact intimately with the protein at the constriction formed by the "G-A-S belt", the actual selectivity is due to residues at the vestibule. The reviewers agree that this is an important contribution to the field but have raised some concerns, which should be addressed in the revised version.

Essential revisions:

1) The Lennard-Jones parameters for K^+^ and Na^+^ used in the calculations are said to be those that optimize the representation of the cation-carbonyl interactions (subsection “System construction and molecular dynamics simulation”, third paragraph), in the context of the CHARMM22 force field. I deduce, therefore, that the authors have not considered subsequent improvements focused on the representation of cation-carboxyl interactions. These interactions were found to be too strong in standard CHARMM22, and to a greater degree for Na^+^ than K^+^ (see e.g. Luo & Roux, JPC Lett 2010; or Marinelli et al., PNAS 2014). Therefore, I believe it is pertinent to question whether free-energy profiles and DDG values calculated with a 'corrected' set of LJ parameters would be significantly different from those currently presented, and if so, whether they would alter the current conclusions of the theoretical study. For example, does the shape of the free-energy profiles in Figure 2 change? One might expect the profile would be shallower around E18/D21. Does the contribution of the constriction near L7, where the Na+/K^+^ selectivity is said to result from different carbonyl interactions and degree of dehydration, become more prominent, relative to that from the E18/D21?

I should note that the authors do not need to carry out any additional simulations to address this question – the task simply requires that they re-calculate the various probability distributions and ensemble averages involved in the derivation of the analysis currently presented, after introducing a 'weight' for each of the snapshots sampled in the existing MD trajectories. This weight is an exponential factor of the change in potential energy resulting from the modification of the force-field. It should be therefore very feasible to provide alternative plots to those shown in Figure 2BC and Figure 3BC, an alternative DDG values to those currently mentioned in the text – which I believe result from the calculations summarized in Figure 2—figure supplement 1.

2) The authors focus their more detailed experimental and computational analyses on acidic residues on the intracellular side of the pore (E18, D18), and highlight these in the title and Abstract of the article. However, the data presented seems to show that the region near L7 is as influential. For example, according to Figure 4, the L7A mutation makes the pore (slightly) K^+^ selective – consistent with this result, the existing free-energy profile in Figure 2 also shows that the L7 region is as important, if not more. If the new calculations with the 'corrected' forcefield parameters mentioned above confirm that the selectivity of the L7 site is similar, or more prominent, than that of the site near E18/D21, it would be necessary to examine the L7 site further, ideally through the same kind of experimental data as that presented in Figure 4, for the acidic region; that is, do the L7A mutations have a progressive effect on the selectivity, like the E18Q mutations? Free-energy perturbation simulations for this site would also be required to rationalize the experimental result, particularly if single, double and triple mutations indeed have a distinct effect, as seen for the site near E18/D21.

3) Simulations that are consistent with the contribution of the side chains of E18' in determining Na/K selectivity were based on the structure of a non-conducting channel. What is the relevance of such simulations for the active open ASIC1?

4) Since the simulations based on the open chick ASIC1 cannot predict the role of E18 on ion selectivity, a logical consequence is that this crystal structure of the cASIC1/toxin complex does not represent the open conformation of the functional channel; this needs to be discussed.

5) What is the contribution of L14' in setting Na/K selectivity? This should be discussed.

6) Obviously other factors than the highly conserved E18 and D20 play a role in channel ion selectivity among the members of the ENaC/degenerins family, as suggested by ENaC that is highly selective (Na/K selectivity >100). This needs to be discussed.

7) Unnatural amino acid replacement is not perfect because it has been reported that under certain conditions, there is non-specific incorporation (Pless et al. (2014) JGP). Please provide appropriate controls to establish that there is no non-specific at the sites tested in this study.

---

## [Author Response]

Essential revisions:

1) The Lennard-Jones parameters for K^+^ and Na^+^ used in the calculations are said to be those that optimize the representation of the cation-carbonyl interactions (subsection “System construction and molecular dynamics simulation”, third paragraph), in the context of the CHARMM22 force field. I deduce, therefore, that the authors have not considered subsequent improvements focused on the representation of cation-carboxyl interactions. These interactions were found to be too strong in standard CHARMM22, and to a greater degree for Na^+^ than K^+^ (see e.g. Luo & Roux, JPC Lett 2010; or Marinelli et al., PNAS 2014). Therefore, I believe it is pertinent to question whether free-energy profiles and DDG values calculated with a 'corrected' set of LJ parameters would be significantly different from those currently presented, and if so, whether they would alter the current conclusions of the theoretical study. For example, does the shape of the free-energy profiles in Figure 2 change? One might expect the profile would be shallower around E18/D21. Does the contribution of the constriction near L7, where the Na+/K^+^ selectivity is said to result from different carbonyl interactions and degree of dehydration, more prominent, relative to that from the E18/D21?

I should note that the authors do not need to carry out any additional simulations to address this question – the task simply requires that they re-calculate the various probability distributions and ensemble averages involved in the derivation of the analysis currently presented, after introducing a 'weight' for each of the snapshots sampled in the existing MD trajectories. This weight is an exponential factor of the change in potential energy resulting from the modification of the force-field. It should be therefore very feasible to provide alternative plots to those shown in Figure 2BC and Figure 3BC, an alternative DDG values to those currently mentioned in the text – which I believe result from the calculations summarized in Figure 2—figure supplement 1.

We agree that the effect of the ion-carboxylate model requires consideration, given the variability in ion parameters available today, and have added a new Appendix to address the matter. The reviewer suggests computing changes to our results based on a modified Lennard-Jones (LJ) parameter for the Na^+^-COO^-^ interaction that better reproduces osmotic pressure measurements for concentrated sodium acetate solutions. The reference Marinelli et al., PNAS 2014 fitted to 3 M sodium acetate or propionate solutions to report an NBFIX r_min_ value of 3.19 Å in place of the usual CHARMM r_mi_n of 3.12 Å from normal parameter mixing rules (as used in our simulations). In Figure 5 we present a comparison of free energy profiles for Na^+^-acetate interactions in aqueous solution for these and other parameters (revised CHARMM27* using mixing rules to yield r_min_=3.12 Å used in this study; older Charmm27 using mixing rules to yield r_min_=3.06 Å; Marinelli et al. rmin∼3.19 Å; and recent rmin of 3.23 Å from Venable, R.M., et al., 2013. J. Phys. Chem. B. 117:10183-10192), and compare to Quantum Mechanical Car-Parrinello (CPMD) simulations, with details provided in the Appendix. While we do acknowledge there may be some errors in the QM model (as discussed in the Appendix), our conclusion is that the parameter we used in this manuscript is capturing a reasonable degree of binding, compared to the QM calculation, whereas the proposed 3.19 Å, as well as the newer 3.23 Å NBFIX values, appear to be too repulsive. We note that our results for NBFIX values are roughly consistent with those of Marinelli et al., and that we have achieved good convergence to within 0.2 kcal/mol for each profile, with attention to correct treatment for Jacobian in both free energy and KD calculation.

In the Figure 5 caption we include K_D_ estimates to compare to the experimental values cited by Marinelli et al. Our chosen model yields a value of 0.9±0.2 M, which is at the bottom of the cited range, whereas the NBFIX=3.19 Å yields a value of 1.8±0.2 M at the top of that range. However, the experimental literature for K_D_ estimates for sodium acetate vary greatly, depending on the approach. While the values cited by Marinelli et al. are 1.17-1.51 M (Ref.#2 of their supporting information), ranges in the literature span an order of magnitude (Fournier et al. 1998. Chem. Geol. 151:69–84). Focusing only on dilute solutions using potentiometry, different experiments have yielded log_10_K_D_ values from 0 – 0.2 (K_D_ of 1.0 – 1.6) at 298K, but the K_D_ drops below 1 for T above 330K (Fournier et al. 1998; noting our simulations for ASIC are done at 323K). Thus, while at the lower end of a range of experiments, our chosen parameter with K_D_ of 0.9±0.2 can be considered to be in approximate agreement with experiments. In support of a lower simulated K_D_ value, we refer to separate non-polarizable and polarizable Amber force field models that yield negative contact free energy minima and computed equilibrium constants (for contact and solvent-separate minima) that equate to overall K_D_ values of 0.70 and 0.85 M (Iskrenova-Tchoukova, E. et al. 2010. Lang- muir. 26:15909–19; Annapureddy, HVR and Dang, LX. 2012. J. Phys. Chem. B. 116:7492−8). Such results for both nonpolarizable and polarizable models in another force field, and with our chosen ion parameter in CHARMM27*, are more consistent with stronger binding.

The reasons for apparent discrepancy in parameters needed to achieve correct interaction or osmotic pressure are interesting. Although non-ideal mixing behavior associated with solute pairing is important, the equilibrium in question is that of equal chemical potential of the solvent on either side of the semi-permeable barrier, for which the solvation of each molecule is also important. i.e. Perhaps one must simultaneously fine-tune the ion and carboxylate hydration free energies alongside the ion-carboxylate interaction. Of course, binding in an ion channel is not the same as an aqueous solution. We have seen in the gramicidin A channel, for instance, that attempting to introduce ion-carbonyl parameters that best fit experimental partitioning data, actually leads to ion energetics in disagreement with experiment (Al- len, TW. et al. 2006. Biophys. Chem. 124:251-267). DE Shaw (Jensen, MØ. et al. 2013. J. Gen. Physiol. 141:619) also demonstrated elimination of knock-on permeation in a K^+^ channel with such repulsive potentials for that ion. They also showed the need for stronger interaction in a channel vestibule to get experimental-like permeation, emphasizing the discrepancy between modeling channel interactions and bulk aqueous media. It may suggest that in a wide vestibular entrance like 4NTW, parameters with stronger interactions with carboxylates are warranted.

We have completed the post-simulation parameter adjustment analysis requested, reported in the new Appendix (see Figure 6 and Figure 7). We note the inherent limitations in such a post-simulation calculation, as it represents a cost of overlapping ion and carboxylate oxygens as they are grown (LJ rmin increased), without relaxing the configurations. In reality such rise in LJ energy would not be seen because the ion and oxygen atoms would move apart, but would be replaced by a change in electrostatic energy. The result of this change of LJ parameter is to add up to ∼0.4 kcal/mol in 4NTW for Na^+^ and K^+^, using the Marinelli et al. increased ion sizes for Na^+^ and K^+^ ions. With similar shifts for both Na^+^ and K^+^ ions, there is no overall significant change in the Na-K free energy difference, keeping the preference for Na^+^ near E18’ and D21’. This remains consistent with our experiments, and our expectations, given the accepted feature that carboxylates favor Na+ binding (e.g. Roux, B. 2010. Biophys. J. 98:2877-2885).

In the case of 2QTS (Figure 6, to compare to the original Figure 3), the use of a more repulsive ion-carbonyl potential raises energies in the narrow closed channel entrance. Deep in the channel’s lower pore, the binding becomes unfavorable, and the estimates unreliable. The approximate post-simulation correction may lead to poor estimates when the perturbations become large (e.g. deep inside a narrowed and more dehydrated pore) where unperturbed configurations deviate from the expected distribution of perturbed configurations. We consider the region near z=-10 Å and above in 2QTS to be unreliable, with increased error and unrepresentative sampling with this correction. However, we see that near D21’ there remains a strong kcal/mol-favoring of Na^+^ over K^+^. Importantly, Na^+^ maintains an entrance

well that is -1 kcal/mol with respect to bulk solution, whereas K^+^ does not.

The requested reweighted analysis for ion complexation is shown in Figure 6 (to compare to Figure 2 and Figure 3), revealing expected reduction in double occupancy with the more repulsive parameters. However, even with this repulsive potential, the probability for tight double pairs remains higher for Na^+^. What is more important, however, is that these multi-ion complex remain energetically favorable for Na^+^ around E18’/D21’. We have also computed adjustments to the Free Energy Perturbation data based on the dominant ion complexes, shown in Figure 7. We maintain a 4NTW free energy difference of 0.2-0.5 kcal/mol for the Na+→K^+^ transformation, and 0.8 kcal/mol for the Na^+^Na^+^→K^+^K^+^ double transformation free energy. In 2QTS, we have a favorable binding of Na^+^ by 1.1 kcal/mol for a single ion and 1.3 kcal/mol in a 2-ion complex, with a net dual ion transformation free energy (Na^+^Na^+^→K^+^K^+^) of 3.0 kcal/mol, favoring Na^+^. These strongly selective relative free energies for ion complexes remain in support of our conclusions, despite the repulsive ion parameters. In summary, we have evidence to suggest the parameters used in our study are appropriate for ion-carboxylate binding. But even with the introduction of a more repulsive parameter, we maintain complex formation and favoring of Na^+^, consistent with our experimental results.

2) The authors focus their more detailed experimental and computational analyses on acidic residues on the intracellular side of the pore (E18, D18), and highlight these in the title and Abstract of the article. However, the data presented seems to show that the region near L7 is as influential. For example, according to Figure 4, the L7A mutation makes the pore (slightly) K^+^ selective – consistent with this result, the existing free-energy profile in Figure 2 also shows that the L7 region is as important, if not more. If the new calculations with the 'corrected' forcefield parameters mentioned above confirm that the selectivity of the L7 site is similar, or more prominent, than that of the site near E18/D21, it would be necessary to examine the L7 site further, ideally through the same kind of experimental data as that presented in Figure 4, for the acidic region; that is, do the L7A mutations have a progressive effect on the selectivity, like the E18Q mutations? Free-energy perturbation simulations for this site would also be required to rationalize the experimental result, particularly if single, double and triple mutations indeed have a distinct effect, as seen for the site near E18/D21.

For better experimental testing of these hypotheses/conclusions, we performed new experiments, measuring the effects on ion selectivity of:

an additional L7’V mutation, reported in a revised Figure 4—figure supplement 1 panel B single, double and triple L7’A concatemers: shown in Figure 4 and Figure 4—figure supplement 1.

Experiment (1) shows that the L7’V substitution, causing essentially a reduction in side chain length/volume, has no ostensible effect on Na^+^/K^+^ selectivity. Changes to L7’ are thus less detrimental to selectivity than at E18’, where both the reduction in length (E18’D) or removal of negative charge (E18’€) abolishes Na^+^/K^+^ selectivity. This new result is described in the first paragraph of the Results section under E18’ determines selective Na^+^ conductance:

“Regarding L7’, it was only substantial changes in side chain size/length that dramatically altered selectivity, with L7’I and even L7’V substitutions not decreasing Na^+^/K^+^ selectivity (Figure 4—figure supplement 1).”

Experiment (2) shows that concatameric channels containing one or even two L7’A-mutated subunits do not differ from WT in their Na^+^ selectivity. This new result provides even stronger evidence for the key role of E18’, further down the pore. The results are discussed in the last paragraph of Results, E18’ determines selective Na+ conductance:

“In contrast, the L7’A mutation, which abolished selectivity in regular channels, caused no ostensible change in relative permeability in 1/3- or 2/3-mutated channels (Figure 4).”

Additional changes in the Discussionunder Determinants of ion selectivity: “Similarly, the L7’ side chain is oriented toward adjacent helices in PDB:4NTW (Baconguis et al., 2014) and is implicated in gating (Yang et al., 2009). Indeed, in our hands the L7’A mutation altered ASIC1a proton-gating more than any other mutant in the present study (Figure 4—figure supplement 1). L7’ might thus be important for pore conformation, but not by contributing to specific interactions with ions in the pore (consistent with the hydrophobic nature of the side chain).”

In light of the above findings (and the fact the we do not see a stepwise effect on selectivity in the mutant concatameric channels), we have opted to not conduct any further free energy perturbation simulations with regard to L7´, as we believe the existing Umbrella Sampling simulations in this narrow single ion region of the pore already capture the relative energetics (as opposed the E18’/D21’ location where FEP served the special purpose of examining dominant multi-ion configurations seen in long unbiased simulations).

3) Simulations that are consistent with the contribution of the side chains of E18' in determining Na/K selectivity were based on the structure of a non-conducting channel. What is the relevance of such simulations for the active open ASIC1?

Our simulations with the open, conducting channel (PDB:4NTW) did indeed show a role for E18’ in selective Na^+^ conduction (Figure 2), although admittedly less so than when using the closed channel (PDB:2QTS). We think our PDB:2QTS simulations are relevant for the open channel because they are, for now, the only simulations we can perform on a less-splayed channel with better resolution of the N- and C-termini, which are truncated in PDB:4NTW (only ~6 amino acids down- stream of E18’/D21’ and known to contribute to ion conduction). This is outlined in the existing introduction to the use of the closed channel PDB:2QTS under Results, Selective ion binding is strongly determined by lower pore structure: “However, the lower pore of this apparently open chick ASIC1/snake toxin complex (PDB:4NTW) is noticeably splayed (Figure 3), possibly as a consequence of the removal of intracellular N- and C-termini for crystallization (Baconguis et al., 2014). […] In the absence of such an ASIC structure, we examined ion binding to PDB 2QTS (Jasti et al., 2007), a closed-channel chick ASIC1 X-ray structure which resolves an additional 10 lower pore residues and shows a significantly narrower lower channel pore (Figure 3).”

However, to further elaborate on our motivation to pursue this strategy, we have also added the following sentence after the above section:

“Although these simulations cannot inform on conduction, they allowed us to explore Na^+^ and K^+^ interactions in a pore for which M2 helices are less splayed.

4) Since the simulations based on the open chick ASIC1 cannot predict the role of E18 on ion selectivity, a logical consequence is that this crystal structure of the cASIC1/toxin complex does not represent the open conformation of the functional channel; this needs to be discussed.

We appreciate this point, and although we hesitate to completely discard the chick ASIC1/toxin complex as a representative open channel, we do wonder if the absence of intracellular termini in that structure causes it to adopt a non-native conformation. Therefore, we have now added the following to a discussion of E18’ at the lower end of channel pore at the end of the section Discussion, Determinants of ion selectivity:

“Finally, it is conceivable that the intracellular entrance to the open ASIC channel pore would adopt a different conformation in the presence of intracellular N- and C-termini, as is certainly the case in structurally analogous P2X receptors (Hattori and Gouaux, 2012; Mansoor et al., 2016).”

5) What is the contribution of L14' in setting Na/K selectivity? This should be discussed.

We have added the following to the section Discussion, Determinants of ion selectivity:

“We propose that the loss of selectivity in the L14’A mutant could be an indirect effect on Na^+^ conduction, caused by the loss of hydrophobic interactions between L14’ side chains and adjacent helices (illustrated by Baconguis et al., 2014), which are likely retained in the more conservative L14’I mutant. […] Furthermore, a role for the L14’ side chain in maintaining the open channel structure is also consistent with the effects of the L14’C mutation on channel gating (Carattino and Della Vecchia, 2012).”

6) Obviously other factors than the highly conserved E18 and D20 play a role in channel ion selectivity among the members of the ENaC/degenerins family, as suggested by ENaC that is highly selective (Na/K selectivity >100). This needs to be discussed.

We are glad this interesting point has been raised, and now we have added the following to the second-to-last paragraph of the Discussion.

“As ENaCs show substantially greater Na^+^ selectivity than ASICs (e.g. Gründer and Pusch, 2015), however, significant differences must occur between these two examples of ENaC/DEG channels. […] This notion is consistent with greater permeation of organic cations in ASIC (Yang and Palmer 2014) than in ENaC (Kellenberger et al. 2001).

7) Unnatural amino acid replacement is not perfect because it has been reported that under certain conditions, there is non-specific incorporation (Pless et al. (2014) JGP). Please provide appropriate controls to establish that there is no non-specific at the sites tested in this study.

We already provide such controls for the unnatural A11’α construct in in Figure 1, which show that non-specific incorporation is remarkably low, yielding a mere 5 nanoampere of current compared to 6 microampere of current with specific incorporation. However, to be even clearer on this very important point, we have added to Figure 1—figure supplement 1 (which is referred to in the Figure 1 legend) an example recording from the experiment that controls for non-specific incorporation of endogenous amino acids to position A11’, where oocytes are injected with all components of the unnatural amino acid system except for the unnatural amino ac- id itself (which allows for non-specific incorporation, if that position is so predisposed).

Furthermore, we have now added to the bottom of panel C of Figure 4—figure supplement 1 the control experiment for the other unnatural substitution (E18’€), addressing non-specific incorporation of endogenous amino acids into position E18’.